# Evaluating the Economic Feasibility of Verjus Production in Texas Vineyards and Wineries

Cassie Marbach, Charlie Hall and Andreea Botezatu *

Department of Horticultural Sciences, Texas A&M University, 400 Bizzell St., College Station, TX 77843, USA; cassie.hutcheson@tamu.edu (C.M.); chall@tamu.edu (C.H.)
* Correspondence: andreea.botezatu@ag.tamu.edu

**Abstract:** This study assessed the economic viability of producing verjus ("green juice") from cluster-thinned grapes. Utilizing the Delphi Method and insights from an expert panel, a comprehensive partial budget model was constructed for vineyards and wineries, focusing on the financial impact of verjus production. Existing vineyards with cluster thinning practices benefited from verjus production. However, vineyards considering cluster thinning solely for verjus may face lower profit margins without a substantial increase in grape harvest prices. Winery operations were also examined, comparing costs of using verjus as an acidifying agent for wine and producing it as a bottled product. Verjus was relatively more expensive than tartaric acid for acidification, but added volume could offset the cost, making it desirable. Additionally, as a standalone product, verjus showed promising profitability, presenting an opportunity for wineries to explore this niche market and expand product offerings. In conclusion, existing vineyards could benefit from verjus production, while wineries could consider using verjus as an acidifying agent or explore its use as an individual product. Careful consideration of costs and market demand is crucial for informed decisions regarding verjus production.

**Keywords:** verjus; acidification; diversification; green juice; cluster-thinned grapes; winery profitability; vineyard profitability; partial budget

## 1. Introduction

This research study focused on potential financial benefits and costs of implementing verjus production in the vineyard and the winery. This involved looking at the economic benefits of using green grapes that would otherwise be a discarded byproduct from the common practice of cluster-thinning (CT) to a product known as verjus at both the winery and vineyard level. The verjus product itself is a simple green juice, similar to lemon juice, made from the under ripe grapes that are thinned off the vine during the wine grape growing season. This juice has health benefits, can be used as a culinary product, and holds potential as an acidifying agent for wine that has high pH [1–6]. The benefits of verjus were explored using an economic approach by building partial budgets that involve verjus production for the winery and vineyard. The additional revenue and costs were calculated from these budgets, and it was determined that verjus production could be beneficial for vineyards already practicing cluster thinning while also being profitable for wineries. Wineries could sell the verjus as an individually bottled product and increase their net income or could acidify their wines with it and mitigate the extra cost by selling a greater volume of wine.

### 1.1. Current Wine Market

In the global wine market, total wine consumption has declined by 3% in 2020 when compared to wine consumption in 2019 [7]. This trend predates the COVID-19 pandemic, but the disruption caused by this world-wide event has significantly increased the direct-to-consumer shipping of wine by 14% in 2021 [7]. A new approach to wine marketing and

continued innovation is necessary in order to meet the changing needs of wine consumers. According to the Wine Market Council [8], 49% of US adults drink wine. Of this 49%, only 14% qualified as high-frequency wine drinkers (drink more than once a week). However, even though statistically only half of the U.S. is a wine consumer, the United States is the largest consumer and importer of wine in the world, along with being the fourth largest producer globally [9,10].

Within the United States, there are several states that produce the majority of U.S. wine including California, Oregon, Washington, and New York [11]. Even though Texas is not one of the top producers of wine, it is the 4th largest consumer of wine in the United States [7,12]. This indicates that there is already a substantial demand for wine in Texas, which could be potentially exploited by local producers. Texas, which will be the primary area of focus for this study, is proving itself to have a successful and growing wine industry. In the past decade, the number of wineries in Texas has grown from 113 in 2005, to 809 active winery permits in 2023. The total economic activity of the Texas wine industry in 2022 was estimated to be $20.35 billion [13]. This economic activity encompasses not only just the total revenue for the industry, but also encompasses the economic activity that is a direct and indirect result of the Texas wine industry functioning and doing business in Texas.

*1.2. Diversification and Innovation in the Wine Industry*

Previous studies have concluded that there are financial benefits for farms that choose to diversify [14–18]. By implementing diversification, farms can become more resilient to changes that come in either economic, environmental, or political forms [14]. In a study focusing on small farms in the US, researchers found that small farms benefited financially from implementing income diversification efforts through off-farm work and agritourism [15]. In South Africa, Bezuidenhout [14], found that due to the increasing struggle to maintain profit levels, local vineyards decided to diversify into citrus production. In this study, Bezuidenhout concluded that diversification from incorporating citrus production with wine grapes does benefit the farmer financially. In another study, researchers found that small farms once again proved to earn greater profits by diversifying into agritourism [16].

The motivations for diversification are often financial but can also stem from a desire to be more environmentally friendly, or to increase sustainability and better their future farming operations [16,17]. There are numerous variables every growing season that can increase or decrease yields and are often difficult to control. These hazards include pests, fungi, disease, late spring freeze, extended heat periods, flooding, and drought [19]. Within Texas, the major area where vineyard and winery tourism takes place is the Texas Hill Country American Viticulture Area (AVA), which faces animal damage and Pierce's Disease as some of the most limiting factors for grape production [20]. Weather can also play a significant variable role, for example in February of 2021, the state of Texas faced its worst winter storm since 1983, affecting the entire state and all vineyard growing areas [21]. There were numerous consequences that vineyards experienced from this devastating storm as the primary and maybe even the secondary buds were damaged so severely that the following spring's bud burst proved to be significantly lower, resulting in lower overall yields, and some even freezing to the ground [22].

The price of the commodity being sold can also play a role in motivating a vineyard owner to diversify their production since it can be such a significant economic indicator of whether a product is a good investment as price has been shown to be one of the top two variables that most affect the economic sustainability of a farm [23]. The price of wine grapes per ton in the US has dropped according to the California Agriculture Statistics Service, with Cabernet Sauvignon falling 23% [24]. These price changes may drive vineyard owners to diversify their income to counteract these decreasing prices [25]. As farmers become more motivated to find alternative means of generating income, the next question is what method of diversification should they implement?

### 1.3. Diversification through Verjus Production

Verjus, also known as "green juice", "unripe grape juice", "koruk" (Turkish), and "Abe ghureh" (Persian), is a juice created from the crushing of the unripe wine grapes that are often the result of cluster thinning [1,3,5]. Verjus is characterized by being a very sour juice with low sugar and high acidity [5,26]. This product is not a recent discovery, but historically, it has been used as a food and medicinal product beginning in the Middle Ages and is still held in high regard in certain countries today such as Iran [26–31].

In Iran and the Middle East, the medicinal benefits of verjus have been held culturally for many years, but only recently scientists have begun to research if the health claims attributed to verjus have scientific proof [31]. Due to Iranian culture valuing verjus so highly as a medical food, verjus' lipid and hypertension controlling ability has been evaluated, along with the claim that it aids digestion and is an antimicrobial agent [27,31–33]. One study found that verjus contains high levels of antioxidants and organic acids and is microbiologically safe due to its self-protection system [34]. Shojaee-Aliabadi et al. (2013), found verjus can be used to reduce blood pressure, lipids, and body weight, showing it can be a good source of natural antioxidants in food. However, in another study, the results show that verjus has no preventive or therapeutic effect in hypercholesterolemia in rabbits, meaning no plasma lipid-lowering effect [32]. There were still very high polyphenol and antioxidative compound levels in verjus making it a plausible health drink [27,34,35].

It is only in recent years however, that verjus has been rediscovered in Western culture [5], and with this rediscovery comes more demand from consumers at supermarkets and restaurants. Verjus has been shown to be a viable and low-cost alternative to acidic salad dressings such as vinegar or lemon juice and has been used as a flavoring agent in more delicate foods [5,26,28,34,36,37]. With verjus' increase in popularity, this is the time for vineyards to diversify their production and take advantage of the grapes they are already thinning for wine production, and use them in a sustainable, profitable way [38].

### 1.4. Cluster Thinning and Verjus Production

In the wine industry, cluster thinning is a common practice that many high-end vineyards already do to increase the quality of the grapes. There are numerous reasons vineyard managers may choose to incorporate cluster thinning, but the goal is to improve the quality of wine grapes, add vine balance, and increase the vigor of the vine. The parameters that can separate a low-quality grape from a high quality one include the soluble solids (SS), titratable acids (TA), pH, phenolics, and anthocyanin concentrations in the grape [39]. Producers are faced with the decision of cluster thinning to better manage these quality parameters or to bypass this age-old practice to increase overall yields. The most common method of measuring if a vine is balanced is the fruit weight to the pruning weight ratio known as the Ravaz index [40,41].

Due to the wide acceptance of the Ravaz index there have been several studies that have tried to determine the appropriate ratio for certain growing regions and cultivars [41–44]. It is commonly believed that a Ravaz index between 5 and 10 is an indicator of a healthy vine, but this ratio needs to be adjusted for the climate and varieties [42]. The aim of any vineyard manager is vine balance, which is the practice of allowing the grapes enough photosynthetic and reproductive area to fully ripen the number of grapes hanging on the vine, often determined by a minimum soluble solids concentration [45]. To achieve vine balance, the practice of hedging and cluster thinning may come into play. There are different timings and severities of cluster thinning which can result in different SS, TA and pH levels at harvest time [39,42,43,46].

Further effects from cluster thinning can include changes in the phenolic composition of the wine grapes, the color intensity, and the flavors or flavor concentrations in the grapes [47,48]. Additionally, cluster thinning has been shown to affect the phenolic and anthocyanin concentrations [47,49]. Volatile compounds, which play a role in the final aroma profile of the wine, can also be affected by the crop load and benefit from cluster thinning [50]. The overall microclimate within the grape canopy is also important for

maintaining quality grapes, and cluster thinning has been shown to help regulate crop level while loosening clusters to reduce the likelihood of rot [49,51,52].

### 1.5. Economics of Cluster Thinning

The improved grape composition that can result from proper CT techniques may often result in the grape grower charging a higher price per ton for the grapes. In 2008, cluster thinning costs ranged from $520 to $650 per hectare if performed manually, or about $220 per hectare if performed by machine [52]. The price increase for grapes might not fully offset this cost. Preszler et al. [53], found that in the production of Riesling, CT decreased yield, but the increase in the aromatic quality caused by CT could only be justified if the base price for the grapes increased 143%. This price increase may seem unattainable, but the possibility of selling the unripe grapes to make verjus could help cover the costs of CT.

There is still very little research on whether the demand for verjus is substantial enough to compensate for the investment wine growers would make to produce it. Due to the increasing consumer demand for salads and healthier foods, verjus' demand may also be on the rise; as well since it has been shown to be a preferred option for acidic salad dressings due to its lack of acetic acid that can impart vinegar and possibly nail polish aromas to the food [28,54]. A study conducted in Chile, found there to be a significant enough demand for verjus from supermarket consumers to make it a financially beneficial product for vineyards and wineries to produce [38]. Yet, despite the proven viability of this product, a robust economic analysis of implementing verjus production for vineyards producing wine grapes including exploration of demand and financial benefit for vineyards has not been performed in the United States. The goal of this study is to determine the financial viability of producing and using verjus by accurately estimating costs to produce it. Results could prove highly useful to vineyards in today's wine market that are looking to increase their economic and environmental sustainability. The specific objectives for this research study include the following:

Research Question 1: Will harvesting unripe grapes for verjus production be profitable for the vineyard?

Research Question 2: Will using verjus as a replacement for commercial acidifying agents be profitable for the winery?

Research Question 3: Will selling verjus as a commercial product be profitable for the winery?

## 2. Methodology

### 2.1. Overview

The design of this experiment consisted of building separate partial budgets to measure the impact of implementing verjus production on a vineyard and using it in the winery. There are several types of budgets that can be used to measure financial costs and benefits of a business such as the whole-farm budget, enterprise budget, and the partial budget. These budgets are designed with the primary purpose of estimating the input costs, fixed costs, returns, and profits for one acre of a typical Texas vineyard and winery [55]. There will be further discussion of the different types of budgets later. The costs for the partial budget were acquired through three representative growers and four winemakers who were able to supply detailed data regarding their different enterprises. These data sets were then implemented into the model to understand the financial effects of the verjus production and use.

### 2.2. Modeling the Typical Vineyard

Before going into the type of model that will be used in this research study, it is important to establish what a model is and how it can be used to better understand a situation. Murthy [56] defines a model as, "A schematic representation of the conception of a system or an act of mimicry or a set of equations, which represents the behavior of a system". A model represents a real-world situation in a quantifiable way which can help in

better understanding how a system works and how to improve overall performance [56]. A model is based on assumptions and observations and is applicable for farmers due to its relative simplicity and ability to be executed using a spreadsheet program such as Microsoft Excel [10,57]. The benefit of model driven farms is the ability to manipulate different inputs to see what possible outcomes and effects such changes have on the farm [10]. The goal of creating the model is to reproduce the different relationships that occur between all the aspects of the farm so that experimentation can be performed systematically. In the process of simulation, the observer predicts how the changes in inputs affect the outputs to better understand the impact one change will have on the whole system [10].

### 2.3. Implementing a Simulation Model

To use a deterministic model, simulation must be performed on the model to mimic the change in circumstances being applied to the system [10]. If the model is an accurate representation of the vineyard, then the results of running the simulation should be accurate in helping farmers make a more informed decision about adding verjus to their production practices. The method for simulating the vineyard system will generally follow the steps and process developed by Strauss [58] for developing a farm-level model. This approach to creating model is performed in a more positivistic nature, which means the model assumes there are very few adjustments to the farming structure taking place during the simulation, and that the goal of the simulation is to answer the question of "what is the likely impact" rather than "what ought to be" [58]. This aligns with the goals of this research project and should be highly beneficial in providing vineyard management with accurate results for adjusting their production practices.

The steps in developing a simulation for an economic problem begin with defining the problem and objectives for the study, doing an in-depth study of the system in question, and developing a mathematical model of it, running the model, experimenting with the model, and finally analyzing and evaluating the results. Following these guidelines, a partial budget simulation will be applied. The type of simulation that will be used in this study is the standard whole-farm budget model to understand as fully as possible the effects of including verjus production on the vineyard. It is important to note, that a disadvantage of using the simulation model is that the compiler of the model must have a very in-depth understanding of the model due to the numerous variables that must be incorporated into it [14]. To counteract this, the role of experts in developing the model becomes important for developing as accurate a model as possible, and the use of a spreadsheet program will be helpful in integrating the numerous variables that need to be accounted for in the simulation [14,57].

### 2.4. Developing the Typical Farm's Budget Model

The following process of developing a typical Texas vineyard and winery is based on the methods used in McCorkle et al. [59,60] and the Strauss [58] method of developing a typical whole-farm model. The first step in this process is collecting farm-level data from representative vineyard owners. Similar approaches have been used for the Food and Agriculture Policy Research Institute's FAPRI Baseline Economic Outlooks [61]. Menghi et al. [62], also used a grower panel to assess the cost of compliance with new legislation for farmers in the EU. This is then followed by the interview process, which will allow each vineyard representative to provide information such as the size of the vineyard, typical grape varietal, production costs, fixed costs, budgeted yield, yield distribution, price per ton of grapes, and expected yield per acre. The benefits of using a vineyard panel process allows the expert opinions of these growers to be discussed in an open manner with other experts which can help ensure the results are as realistic as possible [60].

The consensus building process, also known as the Delphi method, will be the main form of primary research for developing the typical vineyard budget and model, and secondary research will consist of online and previously developed vineyard budgets for growing grapes in the US including the Arkansas Agricultural Experiment Station

Production Budgets for Wine Grapes [63], the Oregon State University Extension Service Vineyard Economics [64], and McCorkle et al. [60] vineyard estimations. The combination of the primary data from the discussions with the wine grower panels and the secondary data from the literature will help establish an accurate budget for this research project.

The use of a sequence of equations, rather than a complex mathematical formula, can make the simulation more applicable for farmers [14]. A spreadsheet program will capture the variables, calculate the relationships that are occurring on the farm, and create the whole-farm budget [57]. To validate the model the previously developed panel of experts will discuss if the budget and model accurately represent the vineyard and its financial performance in the designated areas [57]. The purpose of the budget is to accurately quantify the effects of a change in the farming system, in this case, the diversification into verjus production.

### 2.5. Partial Budget Analysis

For this study, partial budget analysis is the most appropriate as the expenses being affected by verjus production are very specific, and the effects to the overall profit can then be calculated in a straightforward fashion. The goal of partial budget analysis is to determine how a specific change results in increased costs/income and a decrease in certain costs/income [64]. The steps for producing a partial budget are based on the methods found in Alimi and Manyong [65] and Soha [64]. These changes will be analyzed on a per acre and per hectare basis.

After identifying the change that is going to take place, the different costs, yields, and returns must be listed as something affected by this change. There will be both positive returns and negative effects of this implemented change, and these will be seen in the final costs budget. There may be positive changes to the revenue by expanding the enterprise, or there may be negative effects to the revenue due to losses in yield and added costs. At the end of the partial budget, the net effect is determined by subtracting the negative effects from the positive effects. This will reveal whether the change being implemented has an overall positive effect on the enterprise and is worth pursuing.

To calculate whether the changes are positive for the enterprise, the goal will be to see if there is a positive change in net income. Net income (NI) is the result of the total costs (TC) being subtracted from the total returns (TR). As previously discussed, TC consists of variable and fixed costs. However, since this partial budget assumes the fixed costs are unaffected by the implemented change, they can be assumed as 0 in the final calculation. The end formula for determining if the overall benefits of the change are positive is determined by taking the "change in TR" and subtracting the "change in variable costs" resulting in the "change in NI".

### 2.6. The Delphi Method

The consensus building process will still be used for the stages of determining what the typical Texas vineyard and winery costs and returns are. More specifically the Delphi method will be helpful when trying to reach a consensus about what these typical Texas vineyard and winery expenses are. The Delphi method takes into consideration the panel of experts' answers to survey questions, and then uses an iteration technique to help develop a more robust consensus for the answers provided. The method will begin by providing vineyard and winery experts with an open-ended interview asking for their best estimation of the costs that are listed in the partial budget plus any additional costs they determine might be affected by implementing verjus production, followed by what their expected returns typically are for their enterprise. For the vineyard, this will include the expected returns from selling the grapes to the winery on a per acre basis and converted to a per hectare one. For the winery, their expected returns are calculated on a per gallon basis and converted to liters. The next step is summarizing the results and creating average costs and returns, redistributing this summary to the industry personnel and asking if they agree or disagree with this consensus.

### 2.7. The Partial Budget

From reviewing the literature, it appears that on average when a vineyard undergoes cluster thinning (CT) there is a reduction in yield by about 42% which means that each of the vineyard owners, if not already performing CT, would estimate a crop loss of 42% from implementing the practice [66–68]. It was also taken into consideration whether the farmer was CT before implementing verjus production or would be required to begin this practice. The other costs associated with verjus production will be examined from the results discovered through implementing the Delphi method of analysis from industry expert interviews.

## 3. Results

The first budget focuses on the additional expenses incurred by vineyards when practicing cluster thinning and gathering the resulting grapes for making verjus. In the context of this study, it was observed that only a limited number of vineyards in Texas currently implement cluster thinning as part of their production practices. Conversely, it is more prevalent to find vineyards that do not engage in cluster thinning. Consequently, to develop an accurate and valuable budget reflecting the costs associated with harvesting underripe grapes for verjus production, it is crucial to incorporate the expenses related to cluster thinning into the overall budget. By employing the Delphi method to gather insights from industry experts on cluster thinning practices, a hypothetical model budget was created to estimate the cost of harvesting underripe grapes. The costs are calculated using the U.S. dollar.

### 3.1. Cluster Thinning Estimated Costs

Determining the number of grapes to be removed per acre is the initial activity involved in cluster thinning, which also includes the labor and man hours required for grape removal and their subsequent collection in bins for transportation to the winery. To facilitate estimation, we assumed a grape variety with a high yield, such as Sangiovese or Tannat, as the target for this budget analysis. The estimated tonnage of unripe grapes harvested prior to cluster thinning is derived from the assumption that the vineyard produces 17.3 tons per ha (7 tons per acre). The calculation method involved considering that the average cluster weight per vine after fruit set, during the lag phase, is roughly half the final cluster weight. By doubling the weight of the post-fruit set cluster, the grower can determine the desired percentage to be removed from the final weight. Typically, this falls between 40 and 50%, resulting in a final harvest of approximately 8.6–9.9 tons per ha (3.5–4 tons per acre). Removing 50% of the current cluster weight, at a current yield of 3–4 tons per acre, would yield a harvest of 3.7–4.9 tons of unripe grapes per ha (1.5–2 tons per acre).

The price per ton of grapes for the final grape harvest was determined from the industry panel participants as an average of their normal prices resulting in approximately USD 2600 per ton before cluster thinning was implemented. With cluster thinning implemented, most wineries would increase their price per ton to compensate for the yield losses and possible increase in quality. A post-cluster thinning price increase of 40% was used based on the economic model calculations from Preszler et al. [40], resulting in a price of USD 3640. The net returns for the vineyard that cluster thins, sells the unripe grapes, and sells their final grape harvest for an increased price will still result in a negative return of USD 1520 per acre and USD 3756 per ha.

The labor costs were estimated at USD 14.41 per hour, based on the average wage rate for Texas farmworkers and laborers in the crop, nursery, and greenhouse industry, as reported by the U.S. Bureau of Labor Statistics in 2022 (occupation code 45-2092). Through industry interviews, it was determined that it takes an average of 17.25 h for one person to cluster thin one acre of a vineyard (42.6 h per ha). Typically, these 17.25 man-hours are divided among 5–6 individuals, resulting in approximately 3.5 h dedicated to cluster thinning per acre (8.65 per ha). During this time, a tractor is used to collect the harvested grapes from the bins, with the tractor driver earning an average hourly wage of USD 15.09

in the state of Texas. Additionally, there is a runner who assists the driver by gathering the bins of grapes from the harvesters and delivering them to the tractor.

The tractor's estimated costs were for an 85 Hp tractor and referenced from the Department of Agricultural and Consumer Economics University of Illinois 2021 report on "Machine Equipment Costs: Tractors" where the equipment cost considers fuel, labor, and overhead costs [69].

### 3.2. Additional Cluster Thinning Costs

On average, it was determined that 3.5 man-hours are required per acre to remove hail netting by shifting it to one side (8.6 per ha). Additionally, it is essential to account for the extra labor hours needed to estimate the number of clusters to be thinned. This estimation process involves sampling from a minimum of 3% of the vines to obtain an average cluster count per vine, per acre, which is then used to determine the final cluster weight. For vineyards that do not already practice cluster thinning, it is necessary to purchase hand pruners, as they likely rely on machine harvesters. The initial investment cost of these pruners, along with the purchase of harvester bins, was taken into consideration. Lastly, the time required to replace the hail netting is approximately the same as that spent on removing it, resulting in additional labor costs (See Table 1).

**Table 1.** Model Vineyard Budget for Texas Verjus Harvest.

| Process | Quantity/Acre | Unit | Equipment | Labor | Materials | Total | Cost per Hectare Total |
|---|---|---|---|---|---|---|---|
| Hail netting is removed | 3.5 | man-hours | | $14.41 | | $50.44 | $124.63 |
| Survey is performed to determine cluster thinning amount | 0.5 | man-hours | | $14.41 | | $7.21 | $17.80 |
| Grapes are thinned by hand pruning | 17.25 | man-hours | | $14.41 | | $248.57 | $614.24 |
| Grapes are transported to moving tractor | 3.5 | hours | | $14.41 | | $50.44 | $124.63 |
| Tractor driving along row during thinning | 3.5 | hours | $46.20 | $15.09 | | $214.52 | $530.08 |
| Hand pruners for manual thinning | 5 | pruner | | | $12.50 | $62.50 | $154.44 |
| 5 gallon buckets | 5 | buckets | | | $5.00 | $25.00 | $61.78 |
| Replace hail netting | 3.5 | man-hours | | $14.41 | | $50.44 | $124.63 |
| Cleaning equipment | 0.5 | | | $14.41 | | $7.21 | $17.80 |
| total variable costs for verjus | | | | | | $716.30 | $1770.02 |
| Revenue lost due to lower yields | 4 | tons | | | $2600.00 | −$10,400.00 | −$25,698.92 |
| Revenue from final grape harvest | 2 | tons | | | $3640.00 | $7280.00 | $17,989.24 |
| Revenue from unripe grape harvest | 2 | tons | | | $800.00 | $1600.00 | $3953.68 |
| **Net return** [1] | | | | | | **−$1520.00** | **−$3756.00** |

[1] Calculated using the estimate from adjusted yields less revenue lost.

The partial budget analysis for the estimated change in net revenue for producing verjus in a Texas vineyard resulted in a negative change of USD −2592.95. The reduced revenue calculations can be seen in Table 2 with the mode for the stochastic values being used for the final calculation. Standard deviations for these values were calculated using the minimum, maximum, average values, and a sample size of six.

**Table 2.** The effect of adding verjus production and cluster thinning on net revenue for a Texas vineyard on a per acre basis.

| Added Revenue Due to Change | | Added Costs Due to Change | |
|---|---|---|---|
| Unripe grape harvest | $1600 | Cluster thinning costs | $356.65 |
| Final grape harvest | ($374.16) $7280 ($972.42) | Harvesting costs | $646.60 |
| **Reduced costs due to change** | | **Reduced revenue due to change** | |
| | 0 | Reduced yields | $10,400.00 ($1429.04) |
| Increase in net income | $8880 | Decrease in net revenue | $11,403.25 |
| **Change in net revenue** | **−$2523.25** | | |

The values presented are the modes with standard deviations for stochastic components in parentheses.

An overview of the different types of variables used in this model budget are displayed in Table 3. These variables are either stochastic with a triangular distribution, or deterministic. Understanding the factors that determine whether variables are deterministic or stochastic is important when analyzing the costs of activities such as harvesting grapes in Texas. For some variables, such as cluster thinning preparation and labor, we have specific information and can be certain of their values, making them deterministic. Other variables, such as the price per ton for grape harvesting, are subject to variations due to factors such as market conditions and expert opinions. These variables are stochastic because they can take on a range of values within certain limits. For instance, the price per ton for unripe grapes may vary between $500 and $1400, making it stochastic.

**Table 3.** Overview of variables used in the partial budget model for verjus production in Texas vineyards.

| Budget Variables | Type | Source | Distribution |
|---|---|---|---|
| Cluster thinning preparation and labor | Deterministic | Expert panel | - |
| Cluster thinning labor | Deterministic | National statistics | - |
| Tractor costs | Deterministic | University of Illinois Extension [69] | - |
| Materials for thinning | Deterministic | Expert panel | - |
| Non-cluster thinned grape harvest price per ton | Stochastic | Expert panel and survey | Triangular ($1750, $2600, $3500) |
| Cluster thinned grape harvest price per ton | Stochastic | Preszler et al. [40] | Triangular ($2000, $3640, $4316) |
| Unripe grapes price per ton | Stochastic | Expert panel and survey | Triangular ($500, $800, $1400) |

*3.3. Sensitivity Analysis for Harvesting Unripe Grapes*

When assessing the cost estimation for harvesting unripe grapes in Texas, it is important to conduct a sensitivity analysis to gauge the impact of wage rate fluctuations. Such an analysis allows for a comprehensive understanding of how overall costs may be influenced by variations in wage rates, considering both the state's average wage rate and the national average wage rate. Performing a sensitivity analysis provides valuable insights into the economic implications of changes in labor costs. By considering different wage scenarios, it is easier to ascertain the potential cost implications for grape harvesting operations. This analysis is particularly relevant as wage rates often play a significant role in determining the financial viability and profitability of vineyard activities.

The baseline wage rates serve as the reference point for cost analysis, reflecting the prevailing labor market conditions in Texas and the national average (Table 4). The total costs using the baseline rates in Texas for crop thinning and grape gathering are initially reported as USD 716.30 and USD 790.17, respectively. These values provide a foundation

for evaluating the impact of wage rate fluctuations on overall costs. Increasing the wage rates by 10% reveals the potential cost implications of a modest wage hike. In this scenario, the cost per acre for crop thinning and grape gathering rises to USD 763.01 in Texas and USD 844.27 on a national scale. Next, in the wage rate sensitivity, a 15% increment in wage rates is considered. The corresponding costs per acre amount to USD 786.37 in Texas and USD 871.32 nationally. An even more significant wage rate increase of 25% provides insights into the upper bounds of labor costs. In this scenario, the costs per acre escalate to USD 833.08 in Texas and USD 925.42 nationwide.

**Table 4.** Distribution of costs for harvesting unripe grapes at varying wage rates.

| | Cost to Crop Thin and Gather Unripe Grapes per Acre | | Cost per Hectare to Gather Unripe Grapes | |
|---|---|---|---|---|
| **Mean Wage Rate** | **Texas** | **National** | **Texas** | **National** |
| Baseline | $716.30 | $790.17 | $1770.01 | $1952.55 |
| 10% | $763.01 | $844.27 | $1885.44 | $2086.23 |
| 15% | $786.37 | $871.32 | $1943.16 | $4801.64 |
| 25% | $833.08 | $925.42 | $2058.58 | $2286.76 |

*3.4. Winery's Estimated Costs for Producing Verjus as an Acidifying Agent*

The partial budget analysis for using verjus as an acidifying agent in Texas in place of tartaric acid can be seen in Table 5. The tartaric acid costs of USD 75.70 are per 3785 L of wine to acidify it 0.25 units. The increase in final wine volume due to the addition of 10% verjus (378 L) results in added revenue. The revenue was calculated as additional bottles of wine (504 bottles, 750 mL each) multiplied by $13/bottle. The results indicate that even with verjus costing USD 0.30 per liter of wine compared to the USD 0.02 for tartaric acid per liter of wine, there is still a positive net revenue change of USD 5639.40 from the 10% increase in volume of wine available for sale following the verjus addition.

**Table 5.** The effects of using verjus as an acidifying agent in a Texas winery for 3785 L of wine.

| **Added Revenue Due to Change** | | **Added Costs Due to Change** | |
|---|---|---|---|
| Additional wine volume | $6698 | Unripe grapes | $800.00 |
| | | Production costs | $334.30 |
| **Reduced costs due to change** | | **Reduced revenue due to change** | |
| Tartaric Acid | $75.70 | | $0.00 |
| Increase in net revenue | $6774 | Decrease in net revenue | $1134.30 |
| **Change in net revenue** | **$5639.40** | | |

The second budget (Table 6) focuses on the expenses incurred by the winery in relation to the purchasing of unripe grapes, their pressing, and the utilization of the resulting juice as an acidifying agent in lieu of tartaric acid. The budget table presented in Table 3 provides a comprehensive breakdown of the various costs associated with the winemaking process, including transportation, labor, and materials. The total costs incurred throughout the process amount to USD 1134.30, which represents the financial investment required to produce a quantity of verjus equivalent to 378.5 L. Two initial tests were conducted over the course of two consecutive harvests (2022 and 2023) using different grape varieties (Muscat Canelli and Blanc Du Bois).

**Table 6.** Total costs for producing verjus as acidifying agent in Texas.

| | | | Cost per 378.5 L of Juice | | | |
|---|---|---|---|---|---|---|
| **Process** | **Quantity/378.5 L** | **Unit** | **Equipment** | **Labor** | **Materials** | **Total** |
| Transportation costs using a distributor | 1 truck | | | | | $126.45 |
| Unripe grapes | | ton | | | $800.00 | $800.00 |
| Labor for running forklift and moving grapes into winery | 0.66 | hours | $0.53 | $24.52 | | $16.53 |
| Labor for cleaning equipment before | 1.5 | hours | | $24.52 | | $36.78 |
| KMBS for cleaning | 80 g | grams | | | $1.76 | $1.76 |
| Citric acid powder for cleaning | 198 g | | | | $1.12 | $1.12 |
| Running grapes through destemmer/crusher | 1 | hours | $0.15 | $24.52 | | $24.67 |
| labor and cost to press grapes in bladder press | 2 | hours | $0.15 | $24.52 | | $49.34 |
| labor spent analyzing verjus and trialing wine addition | 0.5 | hours | | $24.52 | | $12.26 |
| labor and materials spent cleaning | 2 | hours | | $24.52 | $2.88 | $51.92 |
| SO2 addition to 40 ppm | | | | | $1.21 | $1.21 |
| labor to acidify wine | 0.5 | hours | | $24.52 | | $12.26 |
| total costs | | | | | | $1134.30 |
| It takes 0.1 L of verjus to acidify 1 L by 0.25 units costing $0.30 cents per liter of wine. | | | | | | |
| It takes 2.5 g of tartaric acid to acidify 1 L by 0.25 units which =$0.02 cents per liter of wine | $234/50 kg of Tartaric acid | | | | | |
| Added revenue from selling 10% more wine | 504 | bottles | | | $13.29 | $6698.16 |

Through these trials, it was determined that an average of 20 mL of verjus per liter of wine was needed to achieve a 0.1 pH unit decrease and that on average 100 mL of verjus per liter of wine were necessary to achieve a 0.25 pH unit decrease. This reduction of 0.25 pH units was identified as the minimum acidification level typically required by most wineries in Texas, as indicated by industry experts. For the purpose of the partial budget analysis, a ratio of 1:10 was adopted, reflecting the amount of verjus needed to acidify the wine by 0.25 units. Additionally, it was observed from previous experiments that approximately 1 ton of unripe grapes yields 378.5 L (approximately 100 gallons) of verjus. Hence, the winery's budget was based on the processing costs associated with 1 ton of unripe grapes, which would produce the 378.5 L of verjus necessary for achieving the minimum desired acidification.

Transportation logistics for grapes from the vineyard to the winery were assessed using the flat rate shipping price model employed by the primary grape distributor in the Texas High Plains area. This model features an even linehaul rate of USD 1000 for transporting grapes from the High Plains AVA to the Texas Hill Country AVA. Given the prevalence of wineries in the Hill Country AVA and the concentration of vineyards in the High Plains AVA, this transportation distance represents a common and applicable scenario for the majority of Texas wineries. To calculate the transportation costs, the fuel rate for the reefer truck was estimated at USD 264.50, resulting in a total freight bill of USD 1264.50. Each reefer truck has the capacity to hold up to 20 tons (40 bins) of grapes. However, it is common for trucks to operate at only half of their maximum capacity during transportation. Therefore, for the baseline estimate, 10 tons of grapes were considered. Based on this estimation, the cost of transporting 1 ton of grapes from the High Plains AVA to the Hill Country AVA amounts to USD 126.45.

Texas wineries, on average, expressed their willingness to pay USD 800 per ton of unripe grapes, while vineyard managers indicated a preference to sell 1 ton of unripe grapes for a price range of USD 500 to USD 1000. To align with the price range desired by wineries and taking into account the average markup of 40% for a ton of ripe grapes [70], $800 can be considered a fair estimate, considering the preferences stated by both parties.

Winery hourly wages were determined using the average weekly wages in wineries, being USD 981 in 2020 from the US Bureau of Labor Statistics and working an average 40-h week, resulting in an average rate of USD 24.52 an hour. The labor involved in running a forklift and transferring the grapes into the winery accounted for 0.66 h and incurred a total cost of USD 16.53. Similarly, the labor required for cleaning the equipment before processing the grapes amounted to 1.5 h and USD 36.78. These tasks are essential for maintaining hygiene and ensuring the quality of the final product. Cleaning materials, such as KMBS (80 g) and citric acid powder (198 g), were utilized during the cleaning process. The costs associated with these materials were USD 1.76 and USD 1.12, respectively. Processing steps, such as running the grapes through a destemmer/crusher and pressing them in a bladder press, required 1 h and 2 h, respectively. Additionally, the analysis of verjus and trialing the verjus addition required 0.5 h of labor, resulting in a cost of USD 12.26. The labor spent cleaning the equipment and storage containers for the verjus is an estimated 2 h, with materials for cleaning totaling USD 2.88. An additional 30 min is needed to complete the acidification process for the wines.

The analysis revealed that the average price for a 50 kg bag of tartaric acid amounted to USD 234. Based on this information, winemakers estimate that it would take approximately 1 g of tartaric acid to acidify 1 L of wine by 0.1 units. Consequently, to achieve an acidification level of 0.25 units, the winemaker would require 2.5 g of tartaric acid. The total cost incurred to acidify 1 L of wine using tartaric acid amounted to $0.02. In contrast, the cost to acidify 1 L of wine using verjus was calculated to be USD 0.30 cents per liter. However, it is important to note that the budget also accounted for the potential impact on sales, considering the possibility of a 10% increase in wine sales due to the larger volume resulting from the addition of the 1:10 ratio of verjus to acidify the wine by 0.25 units. The average price for a bottle of wine was determined to be USD 13.29 in the state of Texas [71]. The verjus additions to the wine would equal an additional 504 (750 mL bottles) of wine to be sold.

The following budget analysis focused on determining the breakeven price required to produce a 750 mL bottle of verjus for commercial sale in Texas, excluding its use as an acidifying agent (see Table 7). A comparison with the budget for creating an acidifying agent replacement shows that the main variations lie in the additional labor, materials, and equipment necessary for bottling the wine, while subtracting the labor requirements for acidifying the wine. Distribution costs were taken into consideration since it is likely that the winery will need to distribute the bottles for commercial purposes.

**Table 7.** Total costs to produce a bottle of verjus in Texas.

| | | | Cost per 378.5 L of Juice | | | |
|---|---|---|---|---|---|---|
| **Process** | **Quantity/378.5 L** | **Unit** | **Equipment** | **Labor** | **Materials** | **Total** |
| Transportation costs using a distributor | 1 | truck | | | | $126.45 |
| Unripe grapes | | ton | | | $800.00 | $800.00 |
| Labor for running forklift and moving grapes into winery | 0.66 | hours | 0.53 | $24.52 | | $16.53 |
| Labor for cleaning equipment before | 1.5 | hours | | $24.52 | | $36.78 |
| KMBS for cleaning | 80 | grams | | | $1.76 | $1.76 |
| Citric acid powder for cleaning | 198 | grams | | | $1.12 | $1.12 |
| Running grapes through destemmer/crusher | 1 | hours | 0.15 | $24.52 | | $24.67 |
| Labor and cost to press grapes in bladder press | 2 | hours | 0.15 | $24.52 | | $49.34 |
| Labor spent analyzing verjus | 0.5 | hours | | $24.52 | | $12.26 |
| Prep, bottling, and cleaning labor costs | 2 | hours | | $24.52 | | $49.04 |
| Mobile bottling line | 504 | bottles | 0.25 | | | $126.00 |
| 750 mL medium weight Bordeaux bottles | 504 | bottles | | | $0.82 | $413.28 |
| Labels | 504 | labels | | | $0.51 | $257.04 |
| DM agglomerate corks | 504 | corks | | | $0.41 | $206.64 |
| Materials for cleaning afterwards | | | | | $2.88 | $2.88 |
| Distribution costs | 500 | miles | $0.29 | $2.50 | | $1395.00 |
| Total costs | | | | | | $3518.793 |
| Breakeven price per bottle = | $6.98 | | | | | |

The mobile bottling line costs were estimated to be USD 0.25 per bottle and did not account for the set-up fee which typically costs USD 1500 because the verjus is likely being bottled at the same time as the winery's other wine products. The materials for the bottling were averaged from the typical bottling materials Texas wineries purchase for their main products. A Bordeaux style bottle that is 750 mL averaged USD 0.82 per bottle, agglomerate corks averaged USD 0.41 per cork, and the labels averaged USD 0.51 per label.

Distribution costs were determined from the average linehaul rate from DAT Freight and Analytics for reefers at USD 2.50 per mile in the Texas region. The fuel surcharge was determined using the Pacific Union surcharge estimator with USD 0.29 per mile added onto the linehaul rate. This was used to estimate a 500-mile trip.

The mobile bottling line costs were estimated to be USD 0.25 per bottle and did not account for the set-up fee which typically costs USD 1500 because the verjus is likely being bottled at the same time as the winery's other wine products. The materials for the bottling were averaged from the typical bottling materials Texas wineries purchase for their main products. A Bordeaux style bottle that is 750 mL averaged USD 0.82 per bottle, agglomerate corks averaged USD 0.41 per cork, and the labels averaged USD 0.51 per label.

The breakeven price for a bottle of verjus produced using a mobile bottling line and distributing a distance up to 500 miles was $6.98 and based on the price per bottle of verjus that has previously been sold in Texas equaling $12 per bottle, the potential profit from bottled verjus was significant.

Distribution costs were highly variable depending on truck type, distance, and demand in the area (see Table 8). Van freight rates were the lowest at USD 2.08 per mile, while reefer freight rates averaged USD 2.50 per mile. Since demand for trucking fluctuates so much, it can be hard to quantify just how much more the linehaul rate would increase in an area with low availability of trucks, but for each reefer truck total, a 30% increase in price was used to determine the cost of distribution with low availability. The fuel surcharge rate according to Union Pacific was USD 0.29 per mile for the average price of diesel fuel of USD 3.50 in Texas.

**Table 8.** Distribution cost variance by truck type, distance, and availability for 1 pallet of verjus.

| | Distribution Costs by Distance | | | |
|---|---|---|---|---|
| Truck Type/Availability | 50 miles | 100 miles | 250 miles | 500 miles |
| Van freight | $118.50 | $237.00 | $592.00 | $1185.00 |
| Reefer freight | $139.50 | $279.00 | $697.50 | $1395.00 |
| Low availability | $181.35 | $362.70 | $906.75 | $1813.50 |

The manual bottling line partial budget estimated that it would take seven people 3 h to produce 42 cases of verjus (504 bottles). With the baseline winery wage rate of USD 24.52, it costs USD 514.92 in labor to produce 504 bottles of verjus. Tables 9 and 10 show the sensitivity analysis of using a mobile bottling line and a manual bottling line with various wage rates and a range of prices for one ton of unripe grapes. The results indicate that the mobile bottling line is less sensitive to an increase in the wage rate.

**Table 9.** Variance in winery production costs per ton of unripe grapes at various wage rates using a manual bottling line.

| | Winery's Cost per Ton of Unripe Grapes | | | | | | |
|---|---|---|---|---|---|---|---|
| Wage Rate | $600 | $680 | $720 | Baseline $800 | $880 | $920 | $1000 |
| Baseline $24.52 | $3546.75 | $3626.75 | $3666.75 | $3746.75 | $3826.75 | $3866.75 | $3946.75 |
| 10% | $3621.93 | $3701.93 | $3741.93 | $3821.93 | $3901.93 | $3941.93 | $4021.93 |
| 15% | $3659.52 | $3739.52 | $3779.52 | $3859.52 | $3939.52 | $3979.52 | $4059.52 |
| 25% | $3734.70 | $3814.70 | $3854.70 | $3934.70 | $4014.70 | $4054.70 | $4134.70 |

**Table 10.** Variance in winery production costs per ton of unripe grapes at various wage rates using a mobile bottling line.

| | Winery's Cost per Ton of Unripe Grapes | | | | | | |
|---|---|---|---|---|---|---|---|
| Wage Rate | $600 | $680 | $720 | Baseline $800 | $880 | $920 | $1000 |
| Baseline $24.52 | $3108.79 | $3188.79 | $3228.79 | $3308.79 | $3388.79 | $3428.79 | $3508.79 |
| 10% | $3127.50 | $3207.58 | $3247.58 | $3327.58 | $3407.58 | $3447.58 | $3527.58 |
| 15% | $3136.97 | $3216.97 | $3256.97 | $3336.97 | $3416.97 | $3456.97 | $3536.97 |
| 25% | $3155.75 | $3235.75 | $3275.75 | $3355.75 | $3435.75 | $3475.75 | $3555.75 |

The variance among the total costs the winery incurred from using a manual versus a mobile bottling line are displayed in Table 11. Without taking into consideration the mobile bottling line set up fee, the manual bottling line was more sensitive to wage rate increases.

**Table 11.** Final costs comparison between mobile and manual bottling lines with various wage rates and baseline price per ton of unripe grapes.

| Winery Wage Rate | Mobile Bottling Line | Manual Bottling Line |
|---|---|---|
| Baseline $24.52 | $3308.79 | $3746.75 |
| +10% | $3327.58 | $3821.93 |
| +15% | $3336.97 | $3859.52 |
| +25% | $3355.75 | $3934.70 |

Lastly, if a winery is an estate winery where they do not have to pay transportation costs, the profit margins for producing verjus are much larger costing only USD 0.17 to acidify 1 L of wine by 0.25 pH units (see Table 12).

**Table 12.** Estate winery's estimated costs to acidify 1 L of wine with 0.1 L of verjus by 0.25 pH units.

| Process | Quantity/378.5 L | Unit | Cost per 378.5 L of Juice | | | |
|---|---|---|---|---|---|---|
| | | | Equipment | Labor | Materials | Total |
| Cost to harvest unripe grapes per ton | 1 | ton | | | | $345.16 |
| Labor for running forklift and moving grapes into winery | 0.66 | hrs | $0.53 | $24.52 | | $16.53 |
| Labor for cleaning equipment before | 1.5 | hrs | | $24.52 | | $36.78 |
| KMBS for cleaning | 80 g | grams | | | $1.76 | $1.76 |
| Citric acid powder for cleaning | 198 g | | | | $1.12 | $1.12 |
| Running grapes through destemmer/crusher | 1 | hrs | $0.15 | $24.52 | | $24.67 |
| Labor and cost to press grapes in bladder press | 2 | hrs | $0.15 | $24.52 | | $49.34 |
| Labor spent analyzing verjus and trialing wine addition | 0.5 | hrs | | $24.52 | | $12.26 |
| Labor and materials spent cleaning | 2 | hrs | | $24.52 | $2.88 | $51.92 |
| Labor to acidify wine | 0.5 | hrs | | $24.52 | | $12.26 |
| SO2 addition to 40 ppm | | | | | $1.21 | $121.00 |
| Total costs | | | | | | $672.80 |
| cost to acidify 1 L of wine with 0.1 L of verjus by 0.25 units = | 0.177755086 | | | | | |

## 4. Discussion

The research objectives for this project were to understand and accurately estimate the steps and costs associated with verjus production both from the vineyard's and the winery's perspective. Additionally, the cost of using verjus as an acidifying agent replacement for tartaric acid was examined. With cluster thinning being a well-known and commonly practiced technique in many areas of the world, verjus provides a use for the otherwise wasted by-product of this grape growing practice [72,73]. Another common problem facing Texas winemakers is high pH grapes and wines, and the cost of acidifying wine down to an acceptable range using tartaric acid, which could be replaced with verjus if economically feasible for the winery [74].

However, it is crucial to assess the economic feasibility of verjus production for both vineyards and wineries, regardless of whether it is used as an acidifying agent or a commercial product. Sustainable production practices must be financially viable for businesses to adopt them, even if the product itself is valuable. Without financial incentives, businesses cannot sustainably engage in the cultivation and production of verjus. Therefore, it is imperative to evaluate the profitability of producing verjus to ensure its long-term viability in the wine industry.

Using the Delphi method, interviews and rounds of communication were conducted to determine what the industry experts in Texas would estimate the additional costs of harvesting the unripe grapes and using them to produce verjus would be. This is a commonly used method to gather opinions and predictions on complex or uncertain issues where there is a lack of definitive knowledge or data [75]. In the context of building a partial budget for producing verjus, the Delphi method was utilized to gather expert opinions on the steps and costs associated with producing verjus in Texas.

Furthermore, it is crucial to consider the potential revenue loss associated with cluster thinning. As highlighted by [53], their model suggests that vineyards would need to raise their price per ton by 143% to compensate for the potential financial losses incurred through cluster thinning. However, an opportunity arises to offset this significant price increase by selling the cluster-thinned grapes. The partial budget model indicates that the vineyard could generate profits ranging from $400 to $500 per ton by selling these grapes. By combining this increased income with a slight markup in the final harvest price per ton, the vineyard could potentially recover the financial costs associated with implementing

cluster thinning. For vineyards that have already integrated cluster thinning into their growing practices, the additional expenses related to gathering unripe grapes can be easily mitigated by the selling price of these grapes.

In regions with high humidity, such as the Texas Hill Country AVA, grape growers may encounter logistical challenges related to Pre-Harvest Intervals (PHI). One common issue affecting these vineyards is black rot, for which Mancozeb is known to be an effective spray [76]. However, it is important to note that Mancozeb has one of the longest PHI durations, requiring a waiting period of 66 days. Since these applications occur early in the growing season, vineyards may face difficulties in maintaining their disease control regimen while also planning to harvest unripe grapes. While alternatives such as using the fungicide Captan later in the season exist, their effectiveness may vary depending on the specific challenges faced by the vineyard. Vineyards located in drier regions such as the High Plains AVA are less likely to be affected by this problem.

### 4.1. Winery Budget Discussion

When evaluating the additional costs associated with verjus production at the winery level, discussions with winemakers revealed that the process itself is straightforward and requires fewer inputs compared to traditional winemaking. However, it is important to note that the cost of each step in the verjus production process can vary significantly among wineries, depending on their size and location. Smaller wineries often have fewer part-time workers, with the winemaker taking on a substantial portion of the workload. In such cases, labor costs may increase as the winemaker may not be receiving the same wage rate as a part-time laborer on average. The size of the winery also impacts equipment availability. Larger wineries typically have an in-house mechanical bottling line, eliminating the need for a mobile bottling line. However, many wineries in Texas are relatively small and could benefit from utilizing a mobile bottling service. Moreover, the distance between the winery and vineyards selling the unripe grapes can pose financial challenges. While the majority of wineries in Texas are located in the Texas Hill Country AVA, the opening of wineries across the state has become more prevalent. Consequently, the transportation costs of grapes can vary significantly depending on the winery's location.

The budget analysis provided data indicating that the cost of acidifying wine with tartaric acid is relatively inexpensive, averaging around USD 0.02 per liter, compared to the cost of using verjus, which averages around USD 0.36 per liter. However, the additional cost of using verjus could be offset by the 10% increase in wine volume obtained through acidification at a 1:10 ratio. Considering the average price of a bottle of wine in Texas at USD 13.29 [71] selling an extra 500 bottles would generate an additional revenue of USD 6645 if the winery were to sell all the wine produced. This translates to an additional revenue of approximately USD 17.72 per liter of wine. With the average cost of producing verjus being USD 3.31 per liter, there is potential to absorb this cost through the sale of more wine.

Nevertheless, there may be challenges related to labeling when acidifying a wine with verjus at a volume exceeding 10%. If the verjus used for acidification is derived from a different grape variety than the wine itself, the wine could be considered a blend if more than 10% of verjus is required to bring the pH to an acceptable range. This situation is likely to occur, particularly with grapes from the Texas High Plains region, which tend to have higher pH levels ranging from 4.2 to 4.3. To lower the pH by 0.50 units and achieve a more acceptable final pH of 3.7, these high pH grapes would likely necessitate acidification with verjus at a 2:10 ratio. This is nearing the legal limit for being able to label a wine as a varietal wine in Texas and the United States, which requires 75% of the wine to be made from a single variety.

There is also the potential to sell the verjus as its own bottled product rather than as an acidifying agent. Even with the extra input costs of a bottling line, bottles, corks, labels, and labor, interviews revealed that a bottle of verjus had previously been sold for USD

12 directly to a restaurant in Texas. This price is almost double the breakeven price for producing a bottle of verjus and is a viable option for wineries to consider diversifying into.

*4.2. Limitations and Future Research*

One limitation of this study is the small sample size of the winemaker and grower panel. The consensus reached by only six experts may not fully represent the diversity of the Texas wine industry, particularly in specific regions such as North Texas or the Bell Mountain AVA, as well as wineries outside the Hill Country AVA. However, it is worth noting that a similar sample size has been successfully utilized in previous studies to develop economic models for previously unexplored situations [59]. Furthermore, there is a lack of vineyards currently implementing cluster thinning practices in Texas. To obtain a more accurate estimate of the time and expertise required for cluster thinning, it would be beneficial to interview growers from regions outside of Texas where cluster thinning is more commonly practiced.

The models presented are only estimations, and no winery will be able to mimic this exact model and costs total. There are so many factors that differ between each winery and vineyard that the model is best seen as a guideline for estimating the actual costs the winery or vineyard would incur when pursuing verjus making.

It is crucial to provide growers with an accurate assessment of the price that winemakers are willing to pay for grapes produced using cluster thinning practices, specifically in the context of Texas grapes. Additionally, it is important to determine whether winemakers can perceive and appreciate the quality differences associated with cluster-thinned grapes. Several studies have highlighted the distinctions between grapes grown with and without cluster thinning [40,43,44,48,50]. However, if Texas winemakers are unable to detect these differences or do not place enough value on them to justify paying a higher price, growers may be reluctant to adopt cluster thinning practices. Therefore, conducting a study that focuses on the ability of Texas winemakers to discern quality differences in grapes and assess a higher monetary value for such grapes would be valuable in determining the economic viability of verjus as an option for growers.

Previous studies have provided limited insights into the sensory characteristics of verjus and how its addition can alter the sensory profile of wine [26,28,54]. However, further research specifically examining the sensory effects of verjus on different wine varieties would be valuable in assisting winemakers in determining its viability as an acidification option. Such a study would provide more specific and detailed information regarding the sensory impact of verjus, aiding winemakers in making informed decisions about its use in wine production.

## 5. Conclusions

Verjus emerges as a versatile and unique product with potential applications in the wine industry. However, the vineyard's budget reveals the challenges associated with its utilization, including the high costs of CT and gathering unripe grapes. Without sufficient price increases or mark-ups, vineyards may struggle to recoup their investments. In addition to the economic considerations discussed, the production of verjus can offer some motivating factors for growers beyond financial gains. Embracing verjus production could enable vineyards to adopt more sustainable practices, which aligns with consumer demand for environmentally friendly products. By utilizing unripe grapes that would otherwise go to waste, vineyards can reduce food waste and maximize resource utilization, fostering a sense of conscious and responsible production. Growers could capitalize on this sustainable practice by marketing verjus as a sustainable and eco-friendly product. This resonates with consumers who prioritize supporting environmentally conscious brands.

The use of verjus as an acidifying agent proves to be more difficult to economically justify for wineries compared to tartaric acid. However, it is favorable to implement its use considering the larger quantity of wine available to sell from the verjus addition, leading to possibly higher sales revenues. It would be valuable for future publications to

investigate the effects of large-scale verjus addition on both red and white wines. If these results demonstrate that verjus acidification has no adverse effects, wineries could benefit from increased revenue by selling a greater volume of wine. When considering verjus as a means of diversifying wine production, data from Texan winemakers suggests that it can be a profitable product when sold as an individually bottled offering. However, it is crucial to ascertain consumers' willingness to pay for such a product to validate the profit potential outlined in this model. Further research is needed in this area. Ultimately, the utilization of verjus in the wine industry warrants continued exploration and evaluation. By understanding its economic implications, production possibilities, and consumer reception, the wine industry can make informed decisions for incorporating verjus into their production practices.

**Author Contributions:** Conceptualization, A.B., C.M. and C.H.; methodology, C.M. and C.H.; validation, A.B., C.M. and C.H.; formal analysis, C.M.; investigation, C.M.; data curation, C.M.; writing—original draft preparation, C.M.; writing—review and editing, C.M., A.B. and C.H.; visualization, C.M., C.H. and A.B.; supervision, C.H.; project administration, C.M.; funding acquisition, A.B. All authors have read and agreed to the published version of the manuscript.

**Funding:** This research was funded by Texas Department of Agriculture, grant number SC-2122-35 and The APC was funded by Enology lab at Texas A&M.

**Acknowledgments:** The wineries and vineyard managers in Texas who contributed their time and expertise. Andrew Lynn, Texas A&M University for the verjus acidification trial results.

**Conflicts of Interest:** The authors declare no conflict of interest.

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
