# Peer review of "Evaluating the Economic Feasibility of Verjus Production in Texas Vineyards and Wineries"

_beverages, doi:10.3390/beverages9030078_

Round 1

Reviewer 1 Report

The authors raise an important issue related to the possibility of using "verjus". However, since this research pertains to a specific area, I suggest tightening the title of the article by adding the phrase “in Texas, U.S.”. I provide more detailed comments below.

I have doubts that a correspondent author may not appear among the co-authors of an article?

I have also some doubts if so long Introduction is needed, however, I suppose this is because the authors wanted to embed the issue of “verjus” very thoroughly in the topic being discussed.

30 cluster thinning > cluster thinning (CT) (The abbreviation “CT” is often used further on, but without explanation so I think it should be explained here since it occurs for the first time.)

42 (Move the title of the subsection to the next line.)

44 consumption [7]. > consumption. (The same reference is placed after the next sentence.)

46 the past year (Instead, I suggest giving a specific year.)

108 AVA (Provide an explanation of the abbreviation in parentheses as it is used for the first time in the text.)

138 et al > et al.

171-172 soluble solids (SS), titratable acidity (TA) > SS, TA (These abbreviations were already explained in the former paragraph.)

182-189 This sentence is too long, rewrite it.

186 grapes. [55] > grapes [55]

272 FARPI (Provide an explanation of the abbreviation.)

272 et al., > et al.

274 EU > European Union (EU)

285 et al., > et al.

296 Move the title of the subsection to the next line.

317-319 I suggest putting the formula instead of the text, because the text is not very understandable.

338 cluster thinning (CT) > CT

Table 1 should be referenced at the beginning of the subsection.

Column headers (Process, Quantity/acre, Unit and Equipment) should be raised by one line.

Add “$” in column “Labor”.

409 costs. (See Table 1) > costs (see Table 1).

421-422 Place Table 2 after this sentence.

422 average. > average (Table 2).

Table 3 should be referenced at the beginning of the subsection.

Column headers (Process, Quantity/378.5 liters and Unit) should be raised by one line.

Add “$” in columns “Equipment” and “Labor”.

441 1134.30 > 1,134.30

502 Move the title of Table 4 to the next line.

Column headers of (Process, Quantity/378.5 liters and Unit) should be raised by one line.

Add “$” in columns “Equipment”, “Labor” and “Materials”.

517 haul linehaul (Is this phrase correct?)

530 refrigerated truck (Perhaps a more appropriate phrase would be “reefer truck” as in line 464.)

545 Table 8 final > Table 8. Final

613 (Beauchamp, 2020) > [73]

Refer to Tables 4-7 at appropriate places in the text.

The layout of all tables (1-8) should be tailored to the requirements of the Journal (see Template), i.e. frames, font, colors and background.

References should be tailored to the requirements of the Journal (see Template), among others: reduce the font size, semicolons between authors, without the “&” sign before the last author, abbreviated journal name and year at the end, before the volume, dates of access to websites, italicized font for Latin names – lines 795, 796, 826, 880, 889).

The attachment of Supplementary Material with tables containing scenarios A and B is unnecessary, as these tables are presented in the text. The table “Hypothetical estate winery” can also be included in the text.

I posted the few comments about English above.

Reviewer 2 Report

there are a lot of questions. the article should be reworked

Round 2

Reviewer 2 Report

in principle, everything is fixed